# Experimental and Numerical Investigation of an Innovative 3DPC Thin-Shell Structure

**Wenfeng Du [1], Liming Zhu [1,*], Hao Zhang [2], Zhiyong Zhou [1], Kewei Wang [1] and Nasim Uddin [3,*]**

[1] College of Civil Engineering & Architecture, Henan University, Kaifeng 475004, China
[2] School of Civil Engineering, Fuzhou University, Fuzhou 350108, China
[3] Department of Civil, Construction and Environmental Engineering, University of Alabama at Birmingham, Birmingham, AL 35294, USA
[*] Correspondence: 10160010@henu.edu.cn (L.Z.); nuddin@uab.edu (N.U.)

**Abstract:** The development and application of new Fiber Reinforced Polymer (FRP) material and 3D printing construction technology provide a basis for making up for the shortcomings of traditional thin-shell structures and building new thin-shell structures with better performance. In this paper, a new 3D Printing Composite (3DPC) thin-shell structure is proposed, which is prepared using a FRP plate as a permanent base mold and combining it with 3D printing cement technology. Both the typical experiment and finite element numerical simulation analysis of the 3DPC thin-shell structure are carried out. The results show that the maximum load capacity of the 3DPC thin-shell structure is increased by 53.3% as compared with the corresponding traditional concrete thin-shell structure. The presence of the FRP sheet effectively delays the generation of initial cracks and enhances the ductility of components.

**Keywords:** composite thin-shell structure; 3D printing; FRP sheet

## 1. Introduction

A thin-shell structure is a kind of curved surface spatial structure with excellent bearing capacity, which has the advantages of a free and beautiful appearance, high stiffness and good integrity. Depending on the rationality of its geometric shape, the axial force is evenly distributed to the shell surface, thus making full use of the compressive performance of the material, so that the structure can cover a long-span space under a few centimeters of thickness [1]. It originated in ancient Rome and was mainly used in religious buildings, such as the Hagia Sophia Cathedral of Istanbul in 998 A.D, St. Peter's Basilica of Rome in 1626, and the Taj Mahal of India in 1629 [2–4]. After the 20th century, with the rapid development of mechanical theories and the widespread application of reinforced concrete materials, new types of thin-shell structures were constantly emerging and widely used in various kinds of buildings, such as the Opera House in Sydney, the National Exhibition Center of Industry and Technologies in Paris, the Beijing Railway Station, the Xochimilco Shell in Mexico City, etc., many of which have become landmarks of a city.

By the late 1970s, with the emergence of new spatial structural systems, the practical application of thin-shell structure had been greatly reduced [5]. It is mainly because more than half of the labour and material resources are wasted in the construction process, during which shell forming needs the erection and removal of temporary formworks and supports. According to the statistical results of engineering examples, almost 60% of the expense of a thin-shell structure is used for construction cost [6]. Another crucial issue is that thin-shell structure is usually sensitive to initial imperfections. The initial defects existing in the long-span thin-shell structure reduce the ultimate bearing capacity greatly in the bearing process, which easily leads to buckling failure [7,8].

However, thin-shell structure is still popular in the architecture field because of its beautiful shape and reasonable bearing performance. Therefore, if new manufacturing

technology or building materials can be applied to improve its shortcomings, the thin-shell structure still has the prospect of broad application.

Additive Manufacturing (AM) is now a promising modern product manufacturing technology and has been commercialized for some decades. It provides new technology, which is expected to overcome the shortcomings of the high construction cost of traditional thin-shell structure. Recent research and practice, such as Contour Crafting [9], D-shape [10] and 3D Concrete Printing [11], have all demonstrated the potential for large-scale processes adopting AM techniques as an alternative way of constructing building components. Providing temporary templates is one of the effective ways to realize form freedom and performance enhancement for 3D printing. Tam et al. [12] proposed the stress line additive manufacturing method to prepare shells, which used a wooden mold as a bottom plate in 2015. Seyedahmadian et al. [13] extruded reinforced polymer composites onto the template in a specific direction to meet the requirements of the structure in 2015. Costanzi et al. [14] presented a new method to print concrete on the temporary freeform surface in 2018. However, the mechanical performance of 3D printed components is found to be susceptible to the removal of the temporary formwork and, as temporary templates usually cannot be reused, the problem of material and cost waste has not been well-solved.

Based on the above summary, an innovative type of a 3DPC thin-shell structure is proposed by using a fiber-reinforced polymer (FRP) plate as a permanent base mold and combining it with 3D printing cement technology in this paper. The 3DPC thin-shell structure takes the FRP sheet as the permanent bottom template, and applies 3D printing cement technology for the reduction of labor costs, which not only solves the problem of cost waste caused by building and removing temporary formworks during the construction process, but also effectively improves the structural integrity and makes full use of the constituent material properties. To fully understand the bearing behavior and failure modes of a 3DPC thin-shell structure, the model tests are carried out and the finite element software ABAQUS is adopted to calculate the nonlinear bearing capacity, which is compared to a corresponding traditional concrete thin-shell structure.

## 2. Experimental Investigations

### 2.1. Experimental Materials

The fiber-reinforced cement material used in the upper layer of the 3DPC thin-shell structure is a new composite material with the advantages of rapid hardening, high early strength and good viscosity. Its main components are Portland cement, silica fume, a water-reducing agent, glass fiber, and water. The polycarboxylate superplasticizer was purchased from Suzhou Xingbang Cooperation (Suzhou, China) as a cement-dispersing agent to improve dispersion of cement particles and reduce the slump loss [15]. In addition, the pre-chopped glass fibers have a length of 12 mm, obtained from the Yongxing Glass Fiber Factory (Yongxing, China). The material properties are provided by the manufacturer. The water used is ordinary tap water. The fiber-reinforced cement material can be obtained through commercialization at present and has been successfully applied to some practical projects, such as the single-story house in Shanghai.

To obtain material parameters of the fiber cement materials, the three-point bending and compression tests were carried out by NEN-EN 196-1 [16] and ASTM C109 [17]. Six samples were tested in each group. To investigate the anisotropy, samples were loaded in X, Y and Z directions and represented by Fx, Fy and Fz. Their mechanical properties were obtained as shown in Table 1. The detailed experimental process is introduced in the literature [18].

**Table 1.** Material parameters of fiber cement materials.

| Material Property | Values |
|---|---|
| Thickness (t) | 10 mm |
| Density ($p_f$) | 2200 kg/m$^3$ |
| Fiber (%) | 1 |
| Longitudinal compressive strength ($F_x$) | 17.34 MPa |
| Transverse compressive strength ($F_y$) | 14.87 MPa |
| Thickness direction compressive strength ($F_z$) | 12.15 MPa |
| Longitudinal flexural strength ($F_x$) | 4.26 MPa |
| Transverse flexural strength ($F_y$) | 7.52 MPa |
| Thickness direction flexural strength ($F_z$) | 9.38 MPa |

The bottom layer of the 2 mm FRP sheet can be directly purchased from the manufacturer, which consists of 70% bidirectional glass fiber wrapped in polypropylene resin. The material properties are shown in Table 2. In addition to light weight and low cost, it also has high durability and fatigue resistance [8], as well as maintenance and bearing capacity. It is more convenient for transportation and construction due to a prefabricated board. Additionally, there is no need for any binder in the preparation process. This is because the fiber-reinforced mortar material has good viscosity, and the bonding force generated in the setting process will make the upper and lower layers closely bonded, thus forming an integrated thin-shell structure [19].

**Table 2.** Material parameters of FRP.

| Material Property | Values |
|---|---|
| Thickness (t) | 2 mm |
| Density ($p_f$) | 980 kg/m$^3$ |
| Fiber (%) | 70 |
| Longitudinal modulus ($E_x$) | 15,169 MPa |
| Transverse modulus ($E_y$) | 15,169 MPa |
| Thickness direction modulus ($E_z$) | 1050 MPa |
| Poisson's ratio ($V_{xy}, V_{yz}, V_{xz}$) | 0.11,0.22,0.22 |
| Shear modulus ($G_{xy}, G_{yz}, G_{xz}$) | 1800 MPa, 1800 MPa, 750 MPa |
| Flexural strength (MPa) | 690 MPa |
| Compressive strength (MPa) | 317 MPa |

*2.2. Sample Preparation*

2.2.1. Constitution of 3DPC Thin-Shell Structure

The 3DPC thin-shell structure with a thickness of 12 mm is prepared in the laboratory and the corresponding detailed dimensions are depicted in Figure 1. Its preparation process is mainly divided into the following two parts: the treatment of the FRP sheet and the preparation of the fiber-reinforced cement surface layer. Different from the traditional manufacturing technology, the 3D printing technology named novel nozzle injection technology is used in the preparation of the fiber cement layer; that is, fibers within the mixed mortar will be aligned along the stress direction through high compression and high shear in the case of the fiber length being larger than the nozzle diameter.

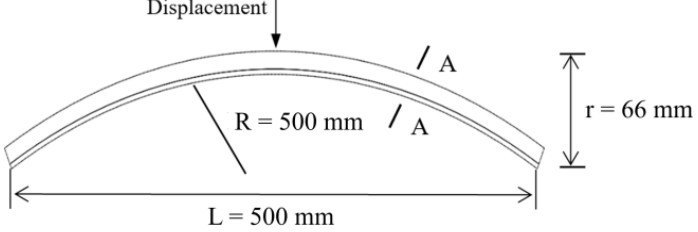

**Figure 1.** The 3DPC thin-shell structure.

Figure 2 depicts the fabrication process of a 3DPC thin-shell structure. First, the short edges of FRP sheet made according to the design curvature were fixed on the self-made scaffold (Figure 2a). Then, the materials prepared according to the mixture ratio were poured into the mixer. After being evenly mixed, the composite mortar was placed in the container of 3D printer HC 1009 (Figure 2b), which is purchased from Huachuang Zhizao Co., Ltd. In the city of Tianjin, China. A large number of studies on materials and printing processes included in product manuals have been carried out by the company. Attention was given to the settings of printing parameters. The horizontal printing speed of the 3D printer was set as 50 mm/s and interlayer interval time was designed as 60 s. Each layer height of 5 mm was adopted, so the machine was designed to move up in 5 mm intervals. Finally, the fiber cement composite was repeatedly extruded along the length direction of thin-shell structure through 10 mm nozzle to cover the FRP sheet (Figure 2c). Note that the excess composite mortar was scraped off along each side of the bottom plate. After being covered with plastic wrap (Figure 2d), specimens were stored for 7 days in the standard curing room. Differing from the traditional composite structure, the preparation process does not require any adhesive. The fiber cement layer and FRP sheet will naturally bond together during the fiber-reinforced cement materials setting process.

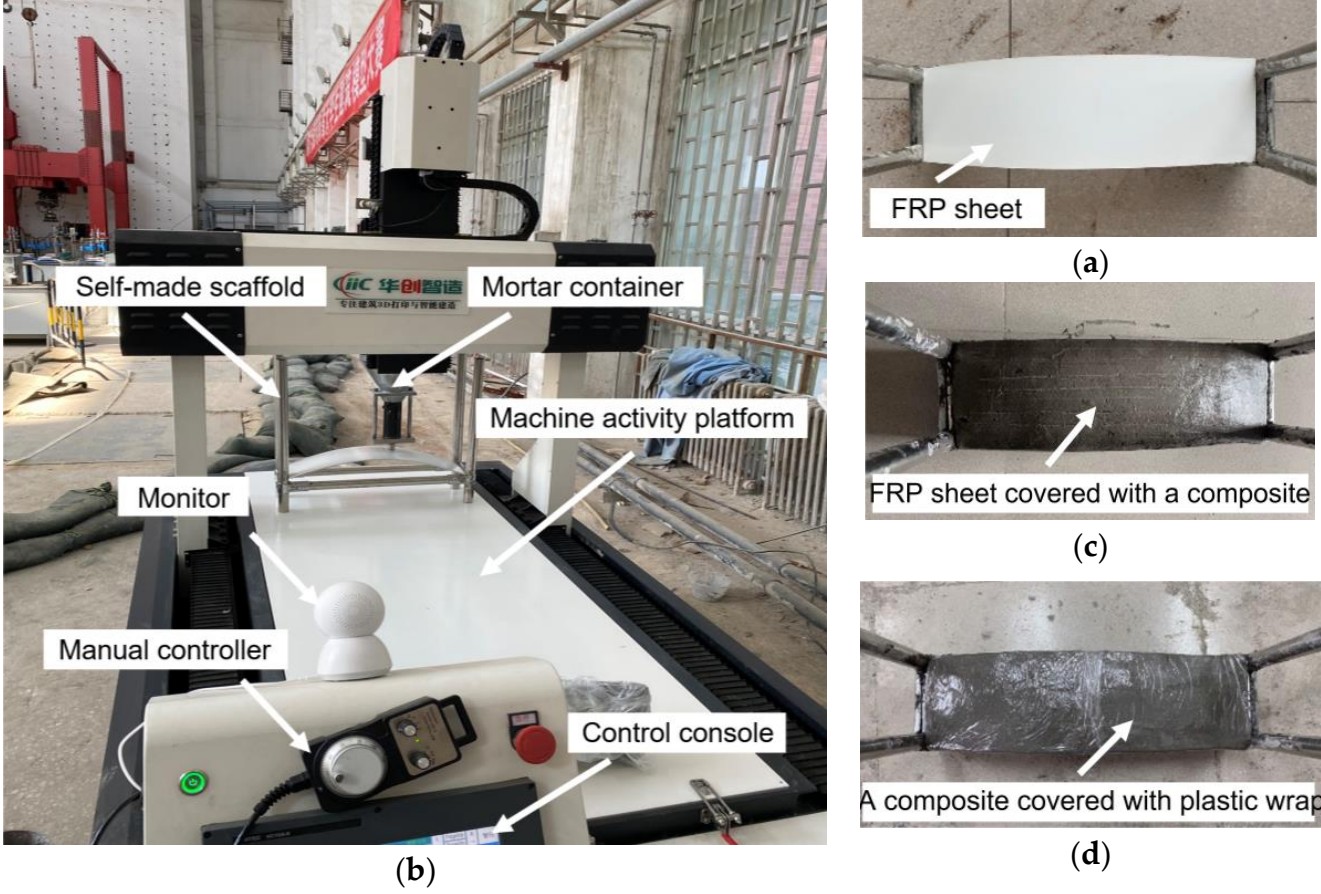

**Figure 2.** Production process of 3DPC thin-shell structure: (**a**) FRP sheet fixed on the self-made scaffold; (**b**) 3D printer HC 1009; (**c**) FRP sheet covered with a composite; (**d**) A composite covered with plastic wrap.

### 2.2.2. Traditional Concrete Thin-Shell Structure

A traditional concrete shell structure with the same thickness was also produced in the laboratory. A galvanized steel wire mesh with a diameter of 2 m, a pitch of 13 mm and yield strength of 300 MPa was adopted as the inner structural steel. The specific dimensions of the traditional thin-shell structure were indicated in Figure 3 and it was fabricated using

traditional production technology (Figure 4). First, the FRP sheet was also fixed to both ends of the self-regulating scaffold. Once fixed, the whole FRP sheet was wrapped with plastic wrap and coated with a proper release agent (Figure 4a). Then, one layer of fresh mixed materials was evenly poured on the FRP panel and the galvanized steel wire mesh with designed size was placed on the surface layer. Finally, the second casting was carried out (Figure 4b). The same as 3DPC thin-shell structure, excess mortar along the bottom edges of FRP template was removed and the specimens were sealed with plastic wrap. After curing for 28 days, FRP template was removed and the traditional concrete thin-shell structure was prepared (Figure 4c).

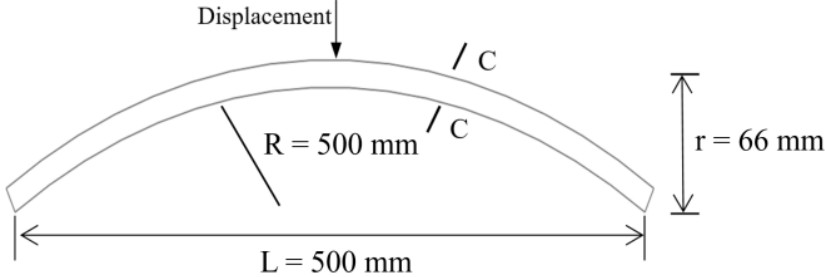

**Figure 3.** The traditional concrete thin-shell structure.

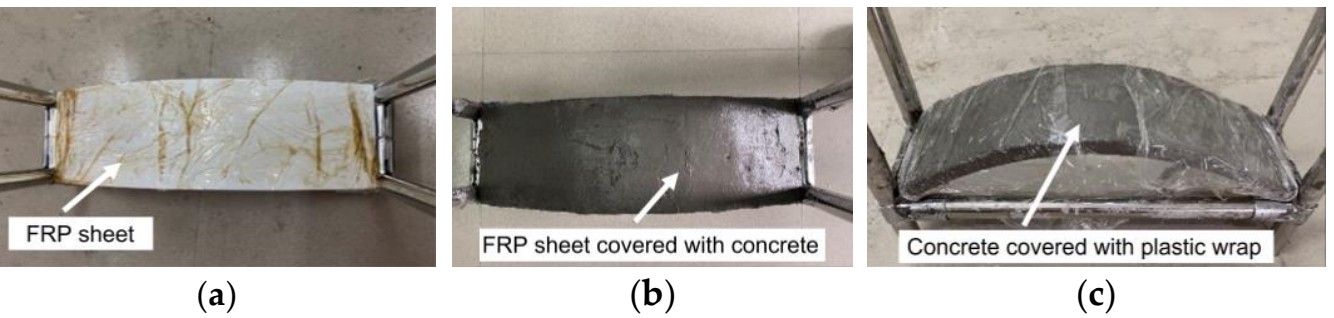

**Figure 4.** Preparation process of traditional concrete shell structure: (**a**) FRP sheet fixed on the self-made scaffold; (**b**) FRP sheet covered with concrete; (**c**) Concrete covered with plastic wrap.

*2.3. Test Setup*

Based on ASTM-C393-00 [20], line loading tests on both 3DPC thin-shell structure and traditional concrete structure were carried out through a Micro Servo-Hydraulic Universal Testing Machine with a maximum load capacity of 300 kN. Three samples were tested for each testing series. As shown in Figure 5, the width directions of the thin-shell structure were fixed at both ends of the self-made scaffold, and the mid-span position was directly under the geometric center of the lower piston. To better realize the line loading test while ensuring the accuracy of test results, a round steel bar with a diameter of 1 cm was customized. Before tests, the pressure-bearing round steel was compacted by the upper dial compact and the interface was coated with a certain amount of lubricant. Thin structures were loaded from 0 kN and a fixed rate of 2 mm/min was set until failure of samples.

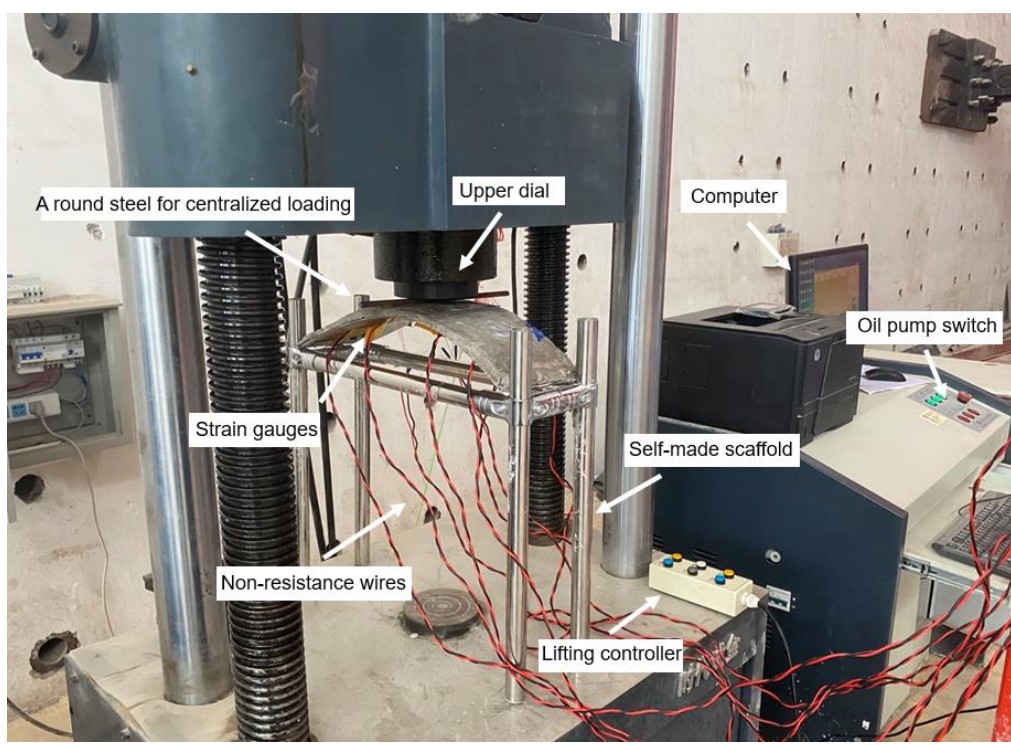

**Figure 5.** Loading test under line loading.

To record the variation state of thin-shell structure under load, four strain gauges were installed on both the structural top and bottom surfaces, which were presented in Figure 6. The type of strain gauge was YEFLA-5 (resistance 120 $\Omega \pm 0.5\%$, sensitivity coefficient $2 \pm 1\%$ and size $5 \times 3$ mm$^2$). Before being placed, the strain gauges were connected to data logger with the non-resistance wires and fixed with insulating glue. The strains of positions were recorded by the data logger controlled by computer.

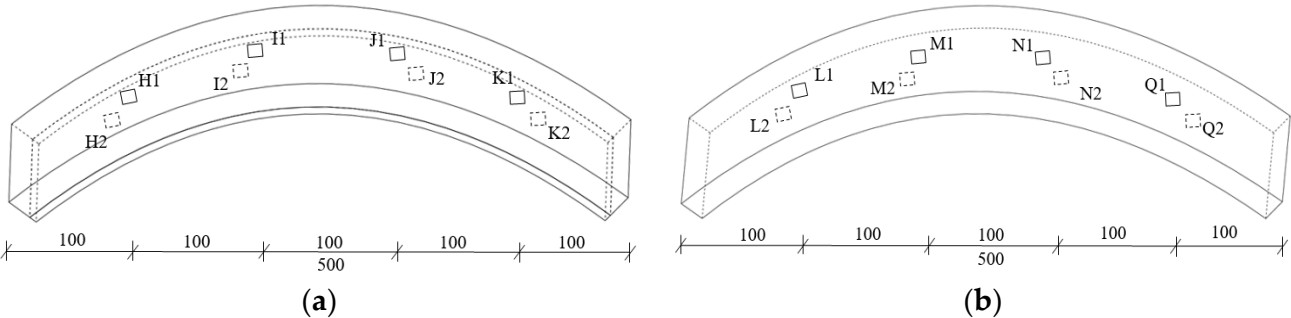

**Figure 6.** Position of strain gauges: (**a**) 3DPC thin-shell structure; (**b**) The traditional concrete shell structure.

## 3. Experimental Results and Discussion

### 3.1. Load-Displacement Behavior

Figure 7 shows the load-displacement curves of 3DPC thin-shell structure and traditional concrete thin-shell structure under line load. From the figure, it can be seen that the load-displacement curve of the traditional concrete thin-shell structure grows linearly before point A at the initial elastic stage. However, with the load increasing, concrete in the tensile zone begins to crack. The structure steps into the nonlinear phase and the tensile force of the structural reinforcement increases. When the mid-span deflection reaches 1.52 mm, the structural maximum bearing capacity reaches 4.36 kN at point D. Subsequently, the bearing capacity shows some gradual drop, while the displacement

continues to increase and cracks expand faster, leading to complete failure of thin-shell structure eventually.

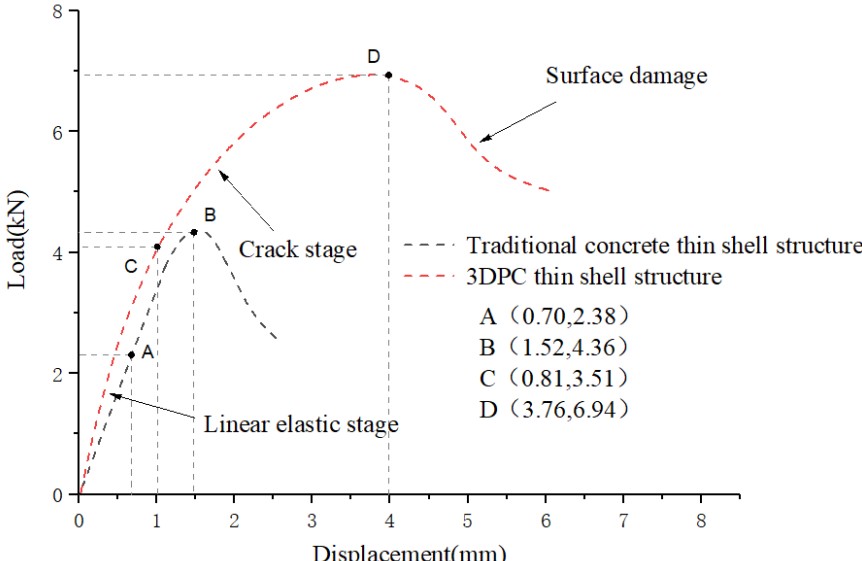

**Figure 7.** Load-displacement curve of the two thin-shell structures.

A 3DPC thin-shell structure, which has more complicated fracture forms, is mainly composed of the linear elastic, crack and surface failure stage. The presence of second and third stages is caused by the debonding of the top and bottom layers and the fracture of the top layer. At the initial stage of displacement loading, there is no obvious deformation, and the relationship between load and displacement increases linearly. The point C corresponds to the end of the linear elastic stage, while the fiber cement layer and the FRP sheet are still under common stress. However, the weak area at the bottom of the top layer tensile zone begins to peel off from the FRP layer and small cracks occur. The load-displacement curve shows nonlinear growth. As the continuous load is increasing, the peak value of the curve appears at point D. The number of cracks at the bottom of the fiber cement surface increases and the width gets larger. Adjacent cracks are connected to form penetrating cracks and the ultimate bearing capacity reaches 6.94 kN. Then, the top layer is destroyed and the curve appears at the descending section. The bearing capacity of the component gradually decreases but still has a certain strength. Additionally, the top layer is completely peeled off from the FRP bottom sheet at the crack position.

*3.2. Load-Strain Behavior*

To evaluate the stress states of the two specimens during the loading process in more detail, eight strain gauges were installed on the top and bottom layers (Figure 6), and the load-strain curves at each position were shown in Figure 8. It can be seen from these figures that the H, I and K positions of the 3DPC thin-shell structure are all under pressure and all curves show obvious linearity in the initial loading stage. Curves slow down gradually until the surface layer is debonding. Figure 8d presents the changing state of the curve at J. At the beginning of loading, the general tendency is the same as that at H, I and K. However, when the load reaches 3.95 kN, the compressive stress condition of J1 is transformed into tensile stress, while J2 is still under pressure. It is the arch occurring at J under load which leads to the debonding of the fiber cement surface layer and FRP sheet, thus changing the stress state of the upper layer.

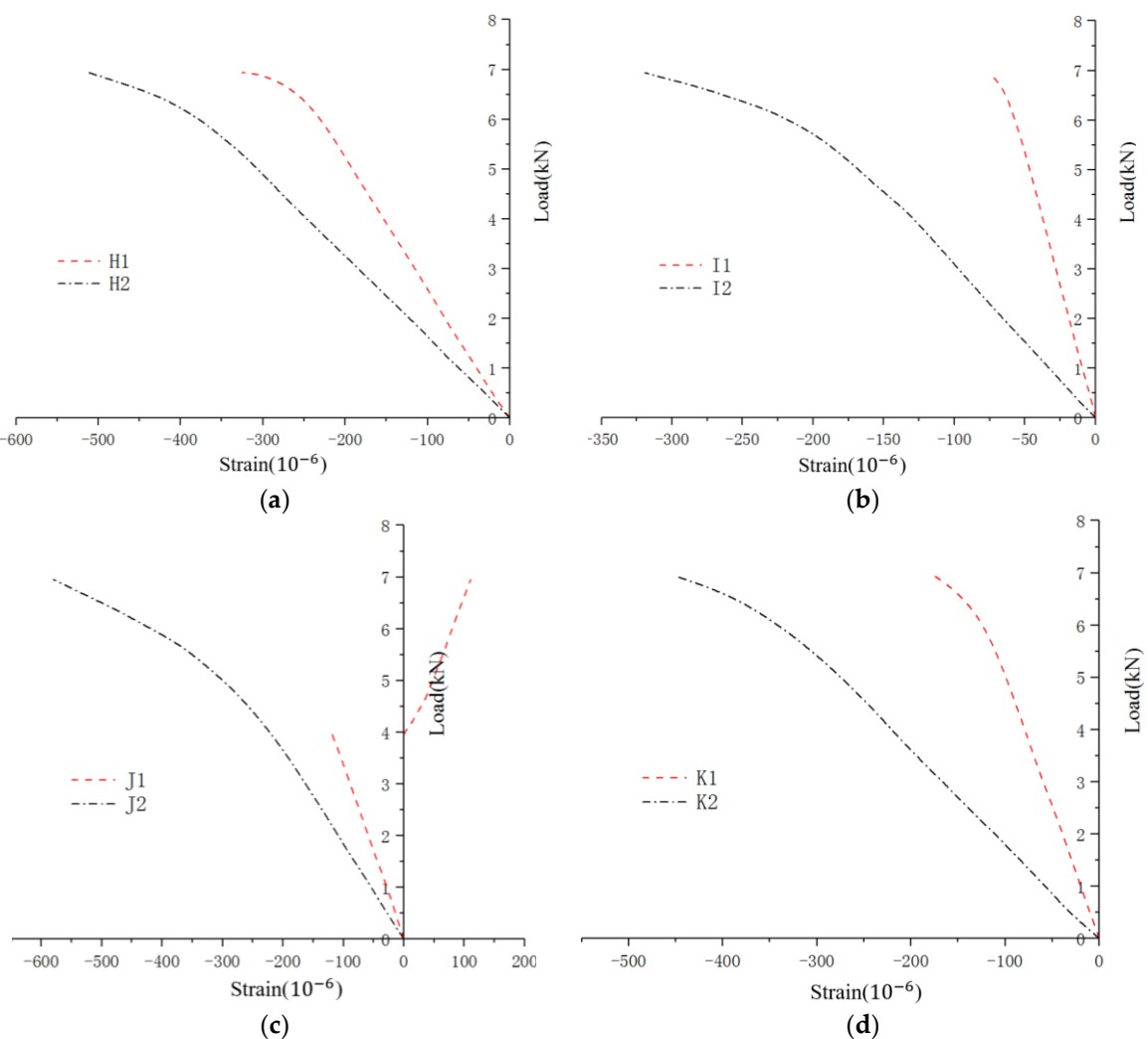

**Figure 8.** Load-strain relationship of the new composite thin-shell: (**a**) Position H; (**b**) Position I; (**c**) Position J; (**d**) Position K.

For the traditional concrete thin-shell structure, the force forms at the corresponding positions of L, M and Q are similar to the 3DPC thin-shell structure. Since position N is the initial damage position of the traditional concrete thin-shell structure, it is chosen as a representative in Figure 9. It can be observed that a continuous change of the force state occurs at position N. Before compressive strain reaches 5.04 μm/m, the load-strain curve at N1 shows a linear increase. Then, the compressive strain gradually decreases and the thin-shell structure changes the force state from compression to tension. The curve starts to increase linearly again until the tensile strain reaches up to 8.87 μm/m. Finally, as load increases to 4.36 kN, the ultimate tensile strain reaches at N1. N2 is still under pressure in the loading process. What is more, the curve enters the nonlinear stage when the load achieves 3.58 kN.

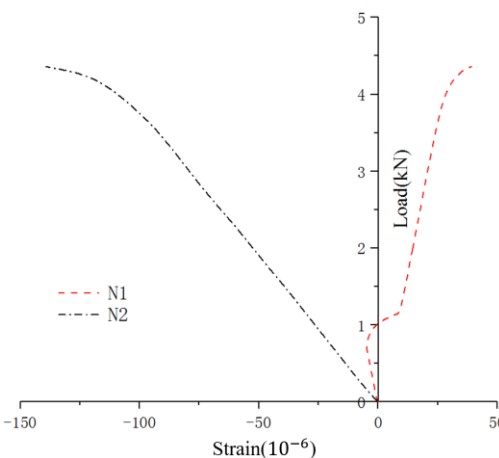

**Figure 9.** Load-strain relationship of the new composite thin-shell in the Position N.

### 3.3. Failure Mode

Figure 10 shows the failure modes of 3DPC thin-shell structure under load. As shown in Figure 10a, the 3DPC thin-shell structure does not show obvious deformation at the initial loading stage and the component is in the elastic stage. With the load increasing, it can be seen that the fiber cement layer and FRP sheet begin to separate at J1 and microscopic cracks appear in the upper layer. The structure enters the nonlinear stage at this time (Figure 10b). When the peak load is reached, the fiber cement layer is damaged. The surface layer at I1 and I2 is slightly depressed downward, but shows no obvious failure form. However, the separation of the fiber cement layer at J1 and FRP sheet takes place, basically resulting in cracks constantly increasing, as shown in Figure 10c. Figure 10d presents the failure mode of thin-shell structure. The loss of structural bearing capacity and tensile failure of the top layer are formed by premature separation of the fiber cement layer at J1 and K1 and FRP sheet.

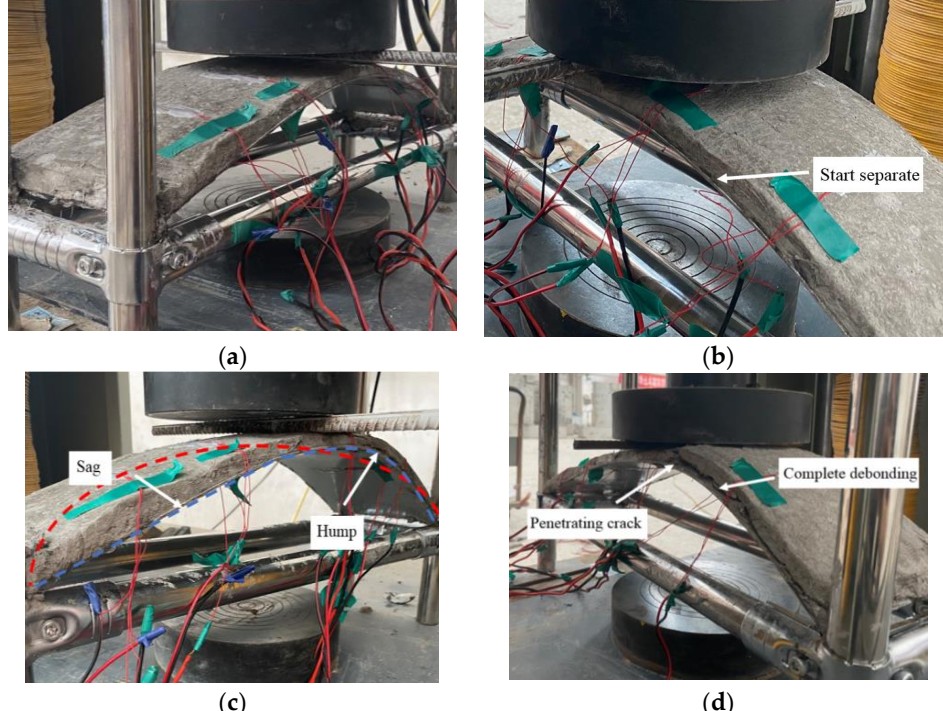

**Figure 10.** Failure pattern of the 3DPC thin-shell structure: (**a**) Elastic stage; (**b**) Nonlinear stage; (**c**) Separation stage; (**d**) Failure stage.

The failure modes of traditional concrete thin-shell structure are shown in Figure 11. It can be found that the bottom of the mid-span and N2 cracks occur, firstly under load, and the number of cracks in the arch increases gradually with an increase in load (Figure 11b). When the maximum bearing capacity is reached, penetrating cracks appear in the arch before mid-span, resulting in structural instability and buckling failure (Figure 11d). It can be seen that the failure form of the traditional concrete thin-shell structure is an asymmetric buckling failure.

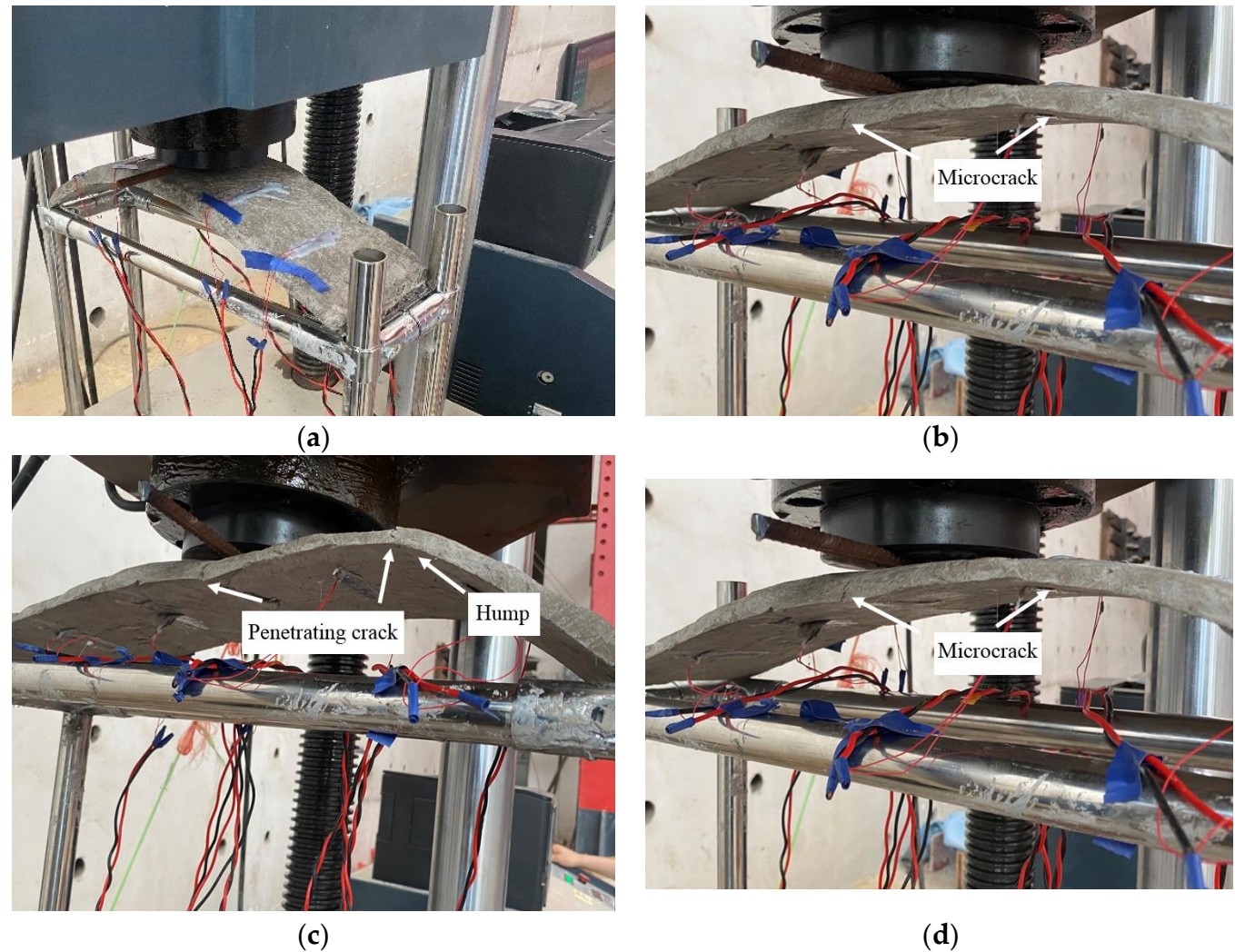

**Figure 11.** Failure pattern of the traditional concrete thin-shell structure: (**a**) Initial loading stage; (**b**) Cracking stage; (**c**) Penetrating stage; (**d**) Failure stage.

Figure 12 shows the penetrating cracks of the upper layer at J1 of 3DPC thin-shell structure. It is clearly seen that there are many cracks in the arch and the fiber distribution is more uniform. Through the partially enlarged view (right), it can be found that the included angle between the fibers and the extrusion direction is small and fibers are basically aligned along the extrusion direction, which proves again that the surface layer prepared by the nozzle extrusion process can obtain a good fiber alignment.

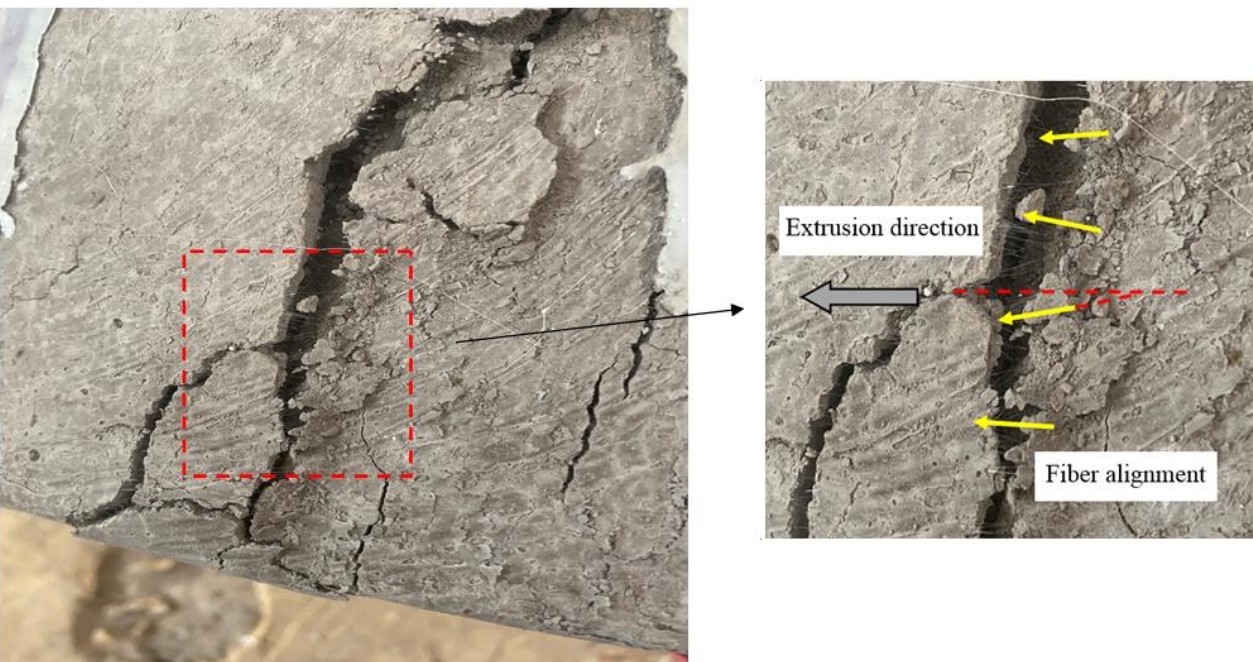

**Figure 12.** Fiber distribution of fracture surface.

### 3.4. Summary of Experimental Results

Based on the experimental results, compared with the traditional concrete thin-shell structure, 3DPC thin-shell structure has reasonable force form, good ductility and stiffness. The results are as follows:

(1) A 3DPC thin-shell structure possesses a reasonable force pattern. It can be seen from the load-displacement curve that the maximum bearing capacity of 3DPC thin-shell structure achieves 6.94 kN and the corresponding mid-span displacement is 3.76 mm, which is 59.2% higher than that of traditional concrete thin-shell structure. In addition, 3DPC thin-shell structure shows better toughness after reaching peak load. It indicates that the combination of FRP sheet and fiber cement materials is conducive to improving the load-bearing capacity and ductility, and promoting the application and development of 3D printing fiber cement technology in the construction field.

(2) The failure modes of two specimens are different. A 3DPC thin-shell structure is damaged in the form of strength failure. Owing to the debonding of surface layers under load, the FRP sheet cannot replace the fiber cement surface layer to bear tensile stress continuedly, causing the upper layer to be damaged by tension. In contrast, the traditional concrete thin-shell structure is buckling failure. During the loading process, the force state of arch is changed from compression to tension, which leads to structural instability and buckling, and specimen failure.

## 4. Numerical Simulation and Comparison Analysis

The finite element analysis of 3DPC thin-shell structure and traditional concrete thin-shell was carried out using finite element software ABAQUS, and comprehensively considered the influence of large deformations, anisotropy, geometric nonlinearity, material nonlinearity response and other factors on the static performance of thin-shell structure. By comparing the results of tests and finite element simulation, the force performance of members was fully analyzed.

### 4.1. Modeling

The thin-shell structure is modeled based on experimental sizes and its specific parameters are shown in Table 3. A 3DPC thin-shell structure contains aligned fiber cement

layers and FRP bottom sheet. Two surface layers utilize an eight-node deformable solid element with linear reduced integration, taking into account the hourglass control and allowable strain. Depending on the measured constitutive relations of aligned fiber cement materials, the monolithic model is applied to simulate fibers dispersed in the cement surface. The separation between FRP sheet and aligned fiber cement layer is simulated by cohesive behavior.

**Table 3.** Dimensions of thin-shell structure.

| Type | Layers | Thickness (mm) | Length (mm) | Hight (mm) | Breadth (mm) |
|---|---|---|---|---|---|
| 3DPC thin-shell | 2 | Upper layer: 12<br>Bottom layer: 2 | 500 | 66 | 100 |
| Traditional concrete shell | 1 | 14 | 500 | 66 | 100 |

For a traditional concrete thin-shell structure, the separated model is applied to model concrete and hot-dip galvanized mesh using different unit types. The corresponding material properties are assigned. A damage plasticity model that uses the concepts of isotropic tensile and compressive plasticity with isotropic damage and C3D8R element which has eight nodes is used for concrete [13]. After package in the assembly module, the hot-dip galvanized mesh is embedded into the concrete element. To simulate the experimental results more realistically, the rigid body of R3D4 element is selected as the pressure bar with a size of $120 \times 6$ mm$^2$ in the thin-shell structure. The center of the pressure bar is chosen as FRP reference point and a displacement controlled loading is applied to constrain X and Y displacement degrees of short sides.

### 4.2. Contact Interactions and Boundary Conditions

There are two ways to simulate the interface bonding in ABAQUS, namely cohesive elements and surface-based cohesive behavior, respectively. Generally, cohesive layers are built by cohesive element in two methods. The first method adopts an overall model and divides the overall grid. A thin layer is cut out on the cross-section where cracks probably appear, defined as cohesive surface. The load and displacement are transferred through common nodes. The second approach models a cohesive layer with 0 thickness using cohesive zone. By assembling, the cohesive layer is placed between the components. The isolated grid is built and connecting interfaces are constrained by ties. Both methods of establishing a cohesive element are complex, so the second method is adopted in this paper.

Surface-based cohesive behavior defines an interaction between interfaces to model the bonding connection without creating a separate unit or model, which is relatively easy to operate. First, the fiber cement layer and FRP bottom sheet are assembled in the assembly module. Then, an interaction property is created, namely cohesive behavior in the interaction module. The cohesive behavior is selected and stiffness coefficient is specified as uncoupled so that all contacting slave nodes use viscous behavior. The stiffnesses in each direction are input. After the secondary damage is added, the damage extension is specified and the damage value is set based on the quadratic tensioning criterion. The evolution type is set as linear displacement. At the same time, the contact property of rigid body is defined. In tangential behavior, the penalty is selected as the friction formula and the friction coefficient value is set as 0.2. In normal behavior, hard contact is selected. Separation after contact is allowed and the rest is kept as default.

The two ends of thin-shell structure are uniformly set as simple supports. The degrees of freedom in the direction of X and Y are constrained, as shown in Figure 13. Since the simulation is a line load test, the reference point is set as the center of the compression bar and the displacement load is applied along direction Y. Other directions are constrained. As the rigid body element, there is no need to couple the reference point with the rigid body before loading and consider that the compression bar is crushed during the loading

process, which improves the calculation convergence and eliminates the stress concentration phenomenon at the loading point.

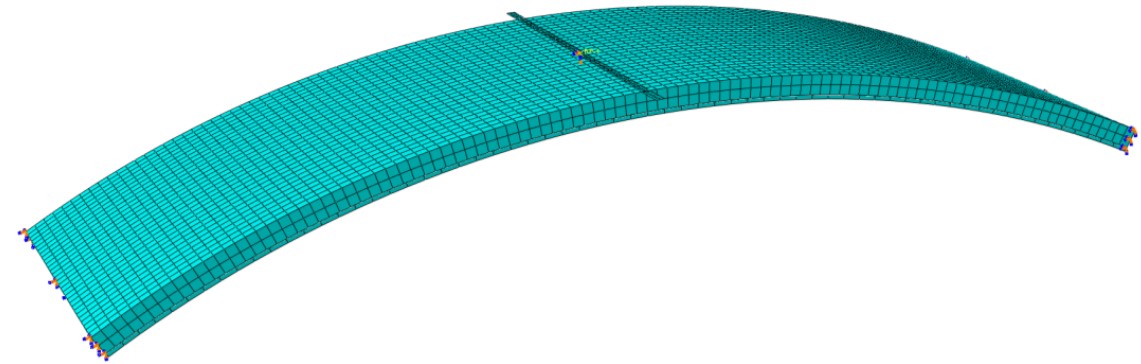

**Figure 13.** Boundary condition and loading for thin-shell structure.

*4.3. Nonlinear Analysis*

Figure 14 indicates the stress distribution of 3DPC thin-shell structure and traditional concrete thin-shell structure at the ultimate bearing capacity. To observe the stress and deformation of components, the scaling coefficient of all cloud images is set as 15. It can be seen from Figure 14a, b that when reaching ultimate bearing capacity, 3DPC thin-shell upper layer has a small compressive stress, only 1.15 MPa, while the mid-span lower layer exhibits the maximum tensile stress of 16.12 MPa, lower than the allowable stress of 690 MPa. This may be because the bottom layer with good elasticity in the tensile weak zone is closely bonded to the top layer, replacing the fiber cement layer to bear tensile stress in the bearing process, so that the component gives full play to its tensile performance.

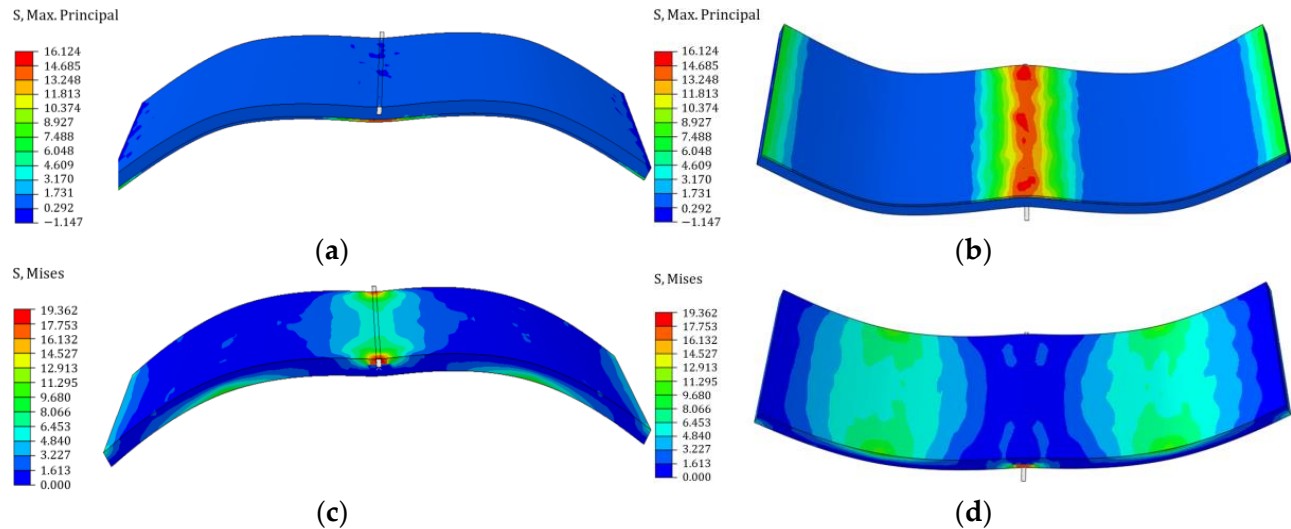

**Figure 14.** Max principal stress visualization: (**a**) Top of 3DPC thin-shell; (**b**) Bottom of 3DPC thin-shell; (**c**) Top of traditional thin-shell; (**d**) Bottom of traditional thin-shell.

Figure 14c, d present the Von Mises stress visualization of traditional concrete thin-shell structure. It can be seen that when the ultimate bearing capacity is reached, the stress distribution of traditional thin concrete shell is not very uniform. M2, N2 and the mid-span upper surface bear great stress of 19.36 MPa, less than the allowable stress of concrete.

Figure 15 demonstrates the load-displacement responses of two thin-shell structures under single-point loading tests. According to the curves, in contrast to the traditional concrete thin-shell structure, 3DPC thin-shell structures with a reinforced FRP sheet have better bearing capacity.

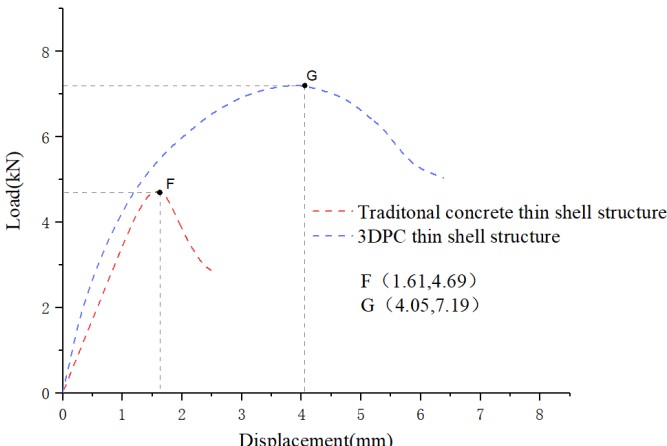

**Figure 15.** Load-displacement curve.

During loading, the bearing capacity of both thin-shell structures increases and the load-deflection curve shows a linear relationship. As load is increasing, the tensile weak area of layer cracks and the traditional concrete thin shell enters the nonlinear stage, while the 3DPC thin-shell structure reinforced by FRP bottom sheet is still in the elastic stage. The traditional concrete thin-shell structure reaches the maximum bearing capacity at F and the displacement is 1.61 mm. For 3DPC thin-shell structure, due to both the FRP bottom sheet and the top layer bearing the stress together under the load, the ductility of the component is greatly improved. Additionally, the component reaches the maximum bearing capacity until load increases to G.

In this paper, the debonding behavior of the 3DPC thin-shell structure layer is simulated by the cohesive behavior of ABAQUS during the loading process. Figure 16 illustrates the displacement program of 3DPC thin-shell structure and the specific peeling situation of the arch position, at almost the middle part of the right half shell. It can be found that the overall shape of thin-shell structure resembles considerably the first-order buckling mode and the arch degree at Y is greater than that at Z. The debonding situations at Z and Y are shown in Figure 16b, c. It can be clearly seen that part of the colloid at Z position has failed, but the top and bottom layers are still bonded to each other. However, the upper and lower layers at Y have been completely peeled off and the stress has exceeded the adhesive stress of the colloid. Thereafter, the FRP sheet will no longer replace the fiber cement layer to bear tensile force, resulting in a large number of cracks occurring in the tensile zone at the bottom of the upper layer and the thin-shell structure fracture.

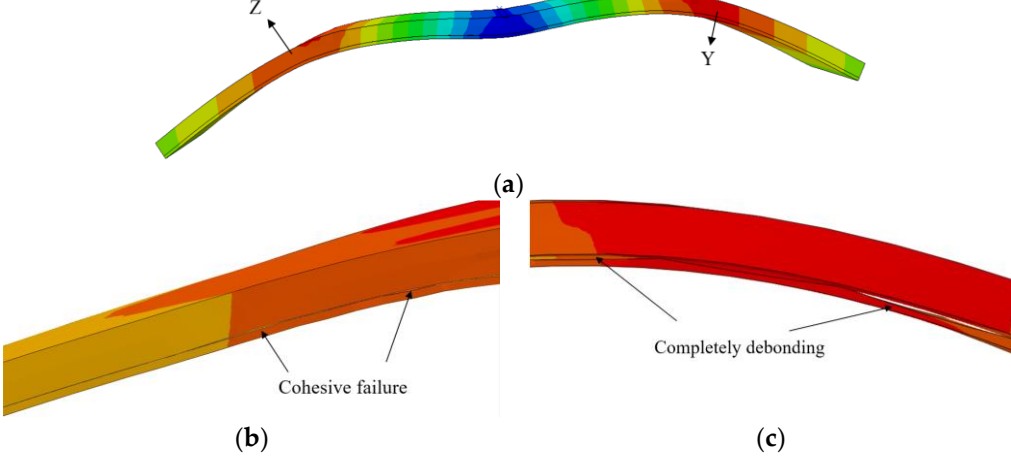

**Figure 16.** Debonding simulation of layers: (**a**) Displacement cloud map; (**b**) Cohesive failure; (**c**) Completely debonding.

### 4.4. Comparison of Experimental and Finite Element Results

Figure 17 displays the comparisons of the simulated maximum bearing capacity of the 3DPC thin-shell structure and traditional concrete thin-shell structure with the experimental results. It reveals that the two curves of 3DPC thin-shell structure exhibit similar variation forms. Compared with the experimental results, the finite element simulation shows better performance. The experimental thin shell reaches the maximum bearing capacity of 6.94 kN at T3 and the corresponding maximum displacement is 3.76 mm, which is 3.6% and 7.7% lower than the finite element simulation results. This may be because the excessive viscosity value set by cohesive behavior exceeds the actual experimental value. When the experimental thin shell is peeling, the upper and lower layers of the simulated thin shell still bear loads together, leading to slow-developing cracks of the bottom fiber-cement layer, and a higher bearing capacity. Additionally, the default bonding strength of the joint surface set in the finite element simulation is uniform and equal, while the FRP sheet and fiber cement layer are not completely bonded tightly in the experiment due to improper operation and environmental conditions. When the layer begins to peel off, the stress at the arch of the fiber-cement layer increases suddenly, leading to an increase in the number of cracks in the tensile zone, thus reducing the ductility of the thin-shell structure. When the layer starts to peel, the stress at the arch of the fiber cement layer increases suddenly, leading to cracks increasing in the tensile zone, which reduces the ductility of thin-shell structure. However, the strength difference between the traditional concrete thin-shell experiment and simulation is because the slip between the structural reinforcement and concrete is not considered in the finite element simulation, resulting in large simulation results. There are more outside factors in the experiment which may affect the member strength.

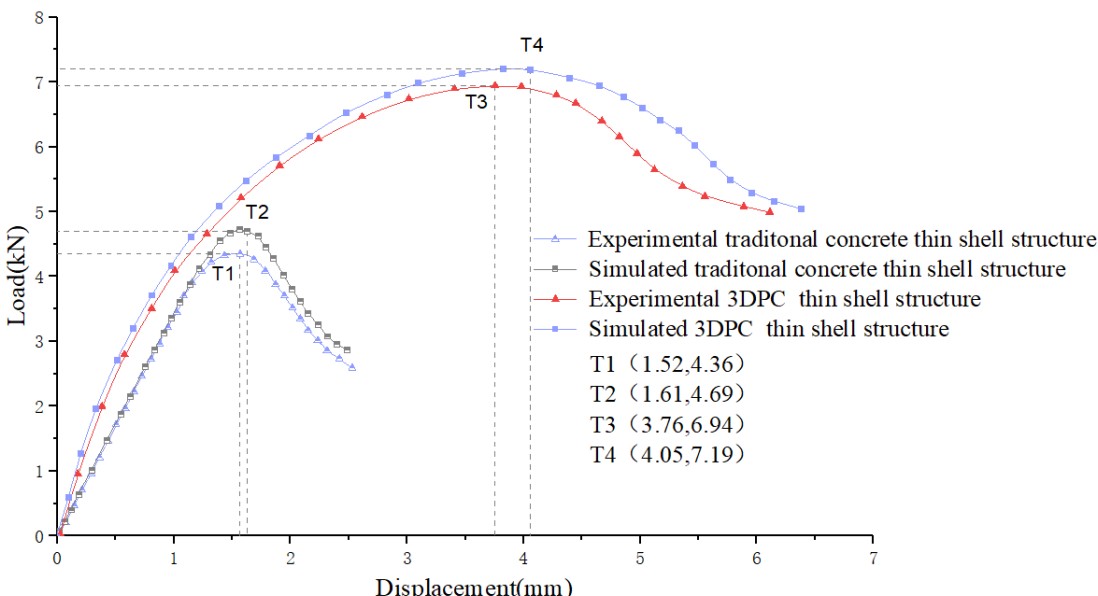

**Figure 17.** The load-displacement curves of the experimental and FEM results.

The results of the finite element simulation and experiment are shown in Figure 18, which shows that both debonding occur at the arch on the right side of the span. It indicates that the simulated surface debonding by cohesive behavior has practical reference value and is consistent well with the experimental phenomenon. It should be noted that in this experiment, due to the uneven bonding of layers, stress concentration occurs at the peeling location, resulting in the formation of penetration cracks and a decrease in ductility compared to the simulated value.

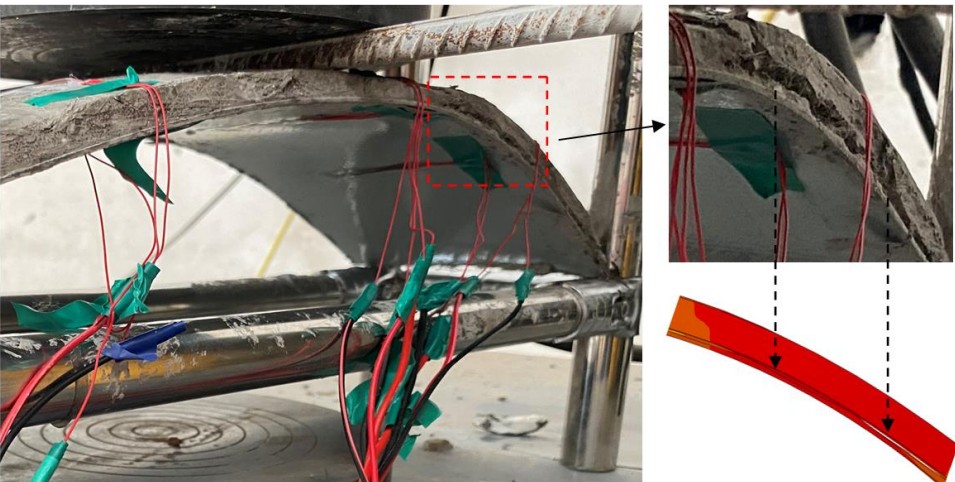

**Figure 18.** Debonding forms of experiments and simulation.

## 5. Conclusions

Aiming at the time-consuming and labor-intensive construction of thin-shell structure, this paper proposes a new 3DPC thin-shell structure. Based on the traditional concrete shell structure, an innovative structure with light weight and high strength is prepared by using an FRP sheet as a permanent base mold combined with 3D printing technology. To fully understand the force performances of components, basic mechanical property tests and finite element simulations are carried out. By analyzing the load-deflection curves, stress-strain curves, stress clouds and failure pattern, the following conclusions are drawn:

(1) A 3DPC thin-shell structure has a reasonable composite constitution. A FRP sheet as the base plate not only solves the biggest formwork construction problem of traditional thin-shell structure, but also provides effective load-bearing capacity and durability for the whole structure. Fiber-reinforced cement material replacing plain concrete can give full play to its compressive performance and the internally aligned fibers improve the ductility of the structure, to a certain extent. The maximum load capacity of a 3DPC thin-shell structure is increased by 53.3% as compared with the corresponding traditional concrete thin-shell structure.

(2) Surface peeling is the main reason for the failure of a 3DPC thin-shell structure. The upper and lower layers are peeled off under load, which gradually increases the tensile stress borne by the fiber cement layer, leading to cracks extending from the bottom to top of the printed layer. The 3DPC thin-shell test results show that when the layer is peeled off, the stress on the top layer of fiber cement material suddenly increased, resulting in cracks and continuous expansion, but layers are not directly pulled off. This is because fibers aligned along the stress direction reduce the stress of materials, delaying the extension of cracks and increasing the ductility of the overall structure.

(3) Based on the experimental data, the simulation of fiber cement layer by using a damage plasticity model is effective and can better reflect the force form of the overall structure. Moreover, the bonding between the fiber-cement layer and a FRP sheet can be well-simulated by surface-based cohesive behavior.

**Author Contributions:** W.D. and N.U. were responsible for the writing of the manuscript and the application of the proposed method. L.Z. and Z.Z. were responsible for the planning of the manuscript. H.Z. and K.W. were responsible for the revision of the manuscript. All authors have read and agreed to the published version of the manuscript.

**Funding:** This research was supported by National Science Foundation in China (NSFC, Grant No. U1704141, 52178172), Henan University Science and technology innovation team support program (Grant No. 22IRTSTHN019), and the Foundation of Zhejiang Provincial Key Laboratory of Space Structures (Grant No. 202106).

**Data Availability Statement:** Some or all data, models, or codes that support the findings of this study are available from the corresponding author upon reasonable request.

**Acknowledgments:** The authors would like to thank Leyu Han, Zhuang Xia, Hui Wang, Yannan Zhao, and Yingqi Wang for their valuable discussion.

**Conflicts of Interest:** The authors declare no conflict of interest.

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
