# Peer review of "Experimental and Numerical Investigation of an Innovative 3DPC Thin-Shell Structure"

_buildings, doi:10.3390/buildings13010233_

Round 1

Reviewer 1 Report

Please find the comments in the attachment.

Author Response

The authors thank gratefully the editor and reviewers for the diligence. On behalf of my co-authors, we would like to express our great appreciation to you.

We have tried our best to improve the manuscript and made some changes to the manuscript. These changes will not influence the content and framework of the paper. We sincerely hope that this revised manuscript has addressed all your comments and suggestions. We appreciate for reviewers’ warm work earnestly and hope that the correction will meet with approval. Should you have any questions, please contact us without hesitation.

Once again, thank you very much for your comments and suggestions.

Yours Sincerely,

Dr. ZHU

Reviewer 2 Report

A new 3D Printing Composite (3DPC) thin shell structure was proposed in this study. Typical tests of 3DPC thin shell structure were carried out and finite element numerical simulation analysis was performed. This topic is worthy of investigations. The following commments are advised to improve the paper.

1. The idea of 3DPC with FRP reinforcement has been proposed by Prof. Feng in Tsinghua University. The authors may cite paper published by Feng et al. 2015 in Composite Structures.
2. How was the concrete surfaced prepared before FRP strengthening?

3. The quality of Figure 2 can be enhanced, in terms of direction and contents. Additional identification words should be added in the figure to highlight the contents which the authors are tended to express. Figures 2b, 2c, 2d fail to show any detail. This is also applicable to Figure 4.

4. What is the reason for involving glass short fibers? Glass fibers are sensitive to pore alkali in concrete.

5. How was the printing direction and stack sequence of the 3DPC shells?

6. The recent studies of FRP reinforced fiber reinforced concrete shells (e.g. Eng Struct 272 2022 115020), which are relevant to this study, should be properly cited and commented.

7. The 3DPC shells included FRP sheet strengthening, while the conventional shells did not include, the comparison between the two types of shells is questionable and should be explained properly.

Author Response

(The authors gave the same response as above.)

Reviewer 3 Report

Comments of the reviewer are included in the files.

Author Response

(The authors gave the same response as above.)

Round 2

Reviewer 1 Report

The paper can be accepted.

Author Response

We have improved English language and style again.

Reviewer 2 Report

Comprehensive revisions should be made before the paper can be accepted.

1. The authors failed to address the comments, expecially for Points 3, 6 and 7.

2. Were the specimens scaled down from a real application? The specimens had very small dimensions and the effect of size is of significance for the behavior of arches. Please elaborate.

Reviewer 3 Report

No further suggestions

Author Response

We have improved the corresponding content.

Round 3

Reviewer 2 Report

The paper in the current form can be accepted.